# The Effect of Leucine-Enriched Essential Amino Acid Supplementation on Anabolic and Catabolic Signaling in Human Skeletal Muscle after Acute Resistance Exercise: A Randomized, Double-Blind, Placebo-Controlled, Parallel-Group Comparison Trial

**DOI:** 10.3390/nu12082421

**Published:** 2020-08-12

**Authors:** Junya Takegaki, Kohei Sase, Jun Yasuda, Daichi Shindo, Hiroyuki Kato, Sakiko Toyoda, Toshiyuki Yamada, Yasushi Shinohara, Satoshi Fujita

**Affiliations:** 1Ritsumeikan Global Innovation Research Organization, Ritsumeikan University, Kusatsu 525-8577, Japan; takegaki@fc.ritsumei.ac.jp; 2Faculty of Sport and Health Science, Ritsumeikan University, Kusatsu 525-8577, Japan; sh0009iv@ed.ritsumei.ac.jp (K.S.); 55fhyanh@gmail.com (J.Y.); ysr15159@fc.ritsumei.ac.jp (Y.S.); 3Ajinomoto Co., Inc., Tokyo 104-8315, Japan; daichi_shindo@ajinomoto.com (D.S.); hiroyuki_kato@ajinomoto.com (H.K.); sakiko_toyoda@ajinomoto.com (S.T.); toshiyuki_yamada@ajinomoto.com (T.Y.)

**Keywords:** LEAAs, resistance exercise, mTORC1, ubiquitin, inflammation

## Abstract

Resistance exercise transiently activates anabolic and catabolic systems in skeletal muscle. Leucine-enriched essential amino acids (LEAAs) are reported to stimulate the muscle anabolic response at a lower dose than whey protein. However, little is known regarding the effect of LEAA supplementation on the resistance exercise-induced responses of the anabolic and catabolic systems. Here, we conducted a randomized, double-blind, placebo-controlled, parallel-group comparison trial to investigate the effect of LEAA supplementation on mechanistic target of rapamycin complex 1 (mTORC1), the ubiquitin–proteasome system and inflammatory cytokines after a single bout of resistance exercise in young men. A total of 20 healthy young male subjects were supplemented with either 5 g of LEAA or placebo, and then they performed 10 reps in three sets of leg extensions and leg curls (70% one-repetition maximum). LEAA supplementation augmented the phosphorylation of mTOR^Ser2448^ (+77.1%, *p* < 0.05), p70S6K^Thr389^ (+1067.4%, *p* < 0.05), rpS6^Ser240/244^ (+171.3%, *p* < 0.05) and 4EBP1^Thr37/46^ (+33.4%, *p* < 0.05) after resistance exercise. However, LEAA supplementation did not change the response of the ubiquitinated proteins, MuRF-1 and Atrogin-1 expression. Additionally, the mRNA expression of IL-1β and IL-6 did not change. These data indicated that LEAA supplementation augments the effect of resistance exercise by enhancing mTORC1 signal activation after exercise.

## 1. Introduction

Skeletal muscle mass is a major determinant of physical performance in humans. Resistance training effectively increases skeletal muscle mass, and numerous nutritional interventions can be used to augment the effect of exercise training [1,2,3,4]. One type of nutritional intervention is supplementation with essential amino acids (EAAs, in particular, leucine). EAAs have a stimulatory effect on muscle anabolism [5,6]. Several previous animal studies have reported that leucine supplementation effectively attenuates skeletal muscle atrophy and/or muscle protein degradation induced by inflammation, disuse, and other factors [7,8]. Thus, EAAs can positively affect not only muscle anabolism but also muscle catabolism.

Resistance exercise is known to transiently activate muscle protein synthesis, and repeated exercise as training accumulates proteins, which leads to skeletal muscle hypertrophy [9,10]. Although the detailed mechanisms are still unclear, the mechanistic target of the rapamycin complex (mTORC) plays a role in these processes [11,12,13]. In particular, mTORC1 was reported to be crucial for skeletal muscle hypertrophy induced by mechanical overload [14], and numerous studies reported the transient activation of mTORC1 after resistance exercise in humans [15,16]. The muscle protein degradation system also regulates skeletal muscle mass, and resistance exercise is known to increase muscle protein degradation [9]. The ubiquitin–proteasome system is one of the major protein degradation systems. In this system, damaged proteins are targeted by ubiquitination and degraded by the 26S proteasome. A previous study reported that resistance exercise acutely increased the mRNA expression of muscle RING-finger protein-1 (MuRF-1)—which is an E3 ubiquitin ligase and promotes protein ubiquitination—in human skeletal muscle [17]. Given that contractile proteins in skeletal muscle, including myosin heavy and light chains and actin, are degraded by this system [18,19,20], the attenuation of the excessive activation of this process may further contribute to increased muscle mass.

EAAs are known to induce muscle anabolic responses, including mTORC1 activation [5,21,22]. Branched chain amino acids (BCAAs; leucine, isoleucine, and valine) stimulate the muscle anabolic response more robustly than other EAAs, and leucine causes the most effective stimulation [23]. However, BCAAs were reported to stimulate mTORC1 signaling more effectively when taken with other EAAs [24]. Based on this information, the intake of EAAs, including leucine at high levels, effectively stimulates the muscle anabolic response, and previous studies reported that leucine-enriched essential amino acids (LEAAs) stimulated the resistance exercise-induced muscle anabolic response in older women at lower levels than whey protein (~30%) [25,26]. However, resistance exercise, especially involving eccentric contraction, causes muscle trauma, and muscle injury is known to activate local inflammation in muscle tissue [27,28]. Additionally, resistance exercise was reported to transiently increase the transcription of inflammatory cytokines [29,30,31], and inflammation is known to enhance proteolytic systems [32,33,34,35]. However, a previous study reported that the intake of BCAAs attenuated the squat exercise-induced increase in serum myoglobin concentration, a marker of muscle damage, in healthy young women [36]. Therefore, researchers proposed that LEAA supplementation not only stimulates the muscle protein anabolic response, but also attenuates the protein degradative response after resistance exercise. However, little is known regarding the effect of LEAA supplementation on the anabolic response in young men, or the response in the degradation system and inflammation after resistance exercise.

In the present study, we investigated the effect of LEAA supplementation on mTORC1, the ubiquitin–proteasome system and inflammatory cytokines after a single bout of resistance exercise in young men. We hypothesized that LEAA supplementation augments mTORC1 activation, and attenuates the increase in ubiquitin ligases and inflammatory cytokines induced by resistance exercise.

## 2. Materials and Methods

### 2.1. Study Design

We performed a parallel-group comparison study between February and June 2019. A total of 24 subjects were recruited; however, 4 subjects dropped out during the study. Thus, the data for 20 subjects were used for the final analysis (10 in the placebo group and 10 in the LEAA group, see Table 1 for the subject demographics). The eligibility criteria were being male and between 20 and 40 years of age. The following exclusion criteria were applied: allergies to soy; habitual resistance exercise training (more than once a week); the habitual intake of protein or amino acid supplements; orthopedic disease or injury in the leg; a history of cardiovascular disease or conditions; the current use of antithrombotic drugs; or other factors identified by the trial supervisor or the trial attending physician. The participants refrained from intense physical activity in the week before the tests. Randomization and blinding (to subjects and those assessing the outcomes) were performed in a random allocation sequence using the envelope method with a 1:1 ratio. Misallocation of one subject in the intervention group was identified after unblinding; however, all data were analyzed as initially randomized according to the intention-to-treat principle. Before participating in the study, the subjects were informed of all procedures and risks, and provided written, informed consent. This study was approved by the Ethics Committee for Human Experiments at Ritsumeikan University (BKC-IRB-2018-058) and the institutional review board of Ajinomoto Co., Inc. (No. 2018-016) and was conducted in accordance with the Declaration of Helsinki. The trial was registered at http://www.umin.ac.jp/ctr/index.htm, with the identifier UMIN000036068. The exercise regimen and data collection were performed at Ritsumeikan University.

### 2.2. Assessment of One-Repetition Maximum

The one-repetition maximum (1-RM) was assessed using the following weight-stack machines: leg extension and leg curl (Life Fitness, Tokyo, Japan). The 1-RM tests were performed based on the procedure recommended by the National Strength & Conditioning Association [37]. Briefly, after the warm-up consisting of one set of 10 repetitions at a level of 40–60% of the estimated 1-RM, and three repetitions at 60–80% 1-RM. Three to four subsequent attempts were performed with progressively increasing weight until the participants failed. The participants took 3 min rests between each attempt.

### 2.3. Study Protocol

The subjects were fed a standard dinner (a boxed meal containing 18.6 g protein, 20.4 g fat, and 120.2 g carbohydrate; the total energy was 758 kcal; consumed at home) and then fasted from 2000 h the previous day. On the morning of the study (0800 h), an indwelling needle was inserted into the median vein in the forearm for venous blood sample collection (Figure 1). After preoperative collection of a blood sample, the first muscle biopsy was obtained from the lateral portion of the randomly selected vastus lateralis muscle. The muscle biopsy was performed using a Bergström muscle biopsy needle (5 mm × 100 mm) under local anesthesia via injection of 1% lidocaine. The collected muscle tissue was immediately rinsed with saline, trimmed, frozen in liquid nitrogen, and stored at −80 °C until analysis. After the first collection of blood samples and biopsy, each subject ingested LEAAs or a placebo with 150 mL of water according to their group (the amount of amino acids contained in the LEAAs was 2.5 g and the compositions of the supplements are shown in Table 2). Immediately after the ingestion, the subjects performed a bout of knee extension and knee flexion resistance exercise (10 reps × three sets) at 70% of the predetermined 1-RM with a 3 min rest between sets, on weight stack machines (Life Fitness, Rosemont, IL, USA). Immediately after the exercise bout, each subject ingested the same supplement again. Seven more sets of blood samples were taken after the completion of the leg extension exercise (Ex), and again immediately and 15, 30, 45, 60 and 90 min after the entire exercise bout. Two more sets of biopsies were taken 10 and 90 min after the exercise bout from the other vastus lateralis muscle. The study schematic is outlined in Figure 1.

### 2.4. Blood Insulin, Glucose, Lactate and Amino Acid Concentrations

The collected blood samples were centrifuged at 1.710× *g* for 10 min at 4 °C, and then serum, plasma and protein-free samples were obtained and stored at −4 °C until analysis. The serum glucose, serum insulin and lactate in the protein-free samples were measured in a clinical laboratory (SRL, Inc., Tokyo, Japan). The plasma amino acid concentrations were analyzed using liquid chromatography–electrospray ionization mass spectrometry (UF-Amino Station, Shimadzu, Kyoto, Japan) followed by precolumn derivatization with APDSTAG^®^ (Fujifilm Wako Pure Chemicals, Osaka, Japan) [38,39].

### 2.5. Western Blotting

The vastus lateralis muscle samples were homogenized and analyzed as described previously, with slight modifications [40]. Briefly, the muscle samples were homogenized in RIPA buffer (Cell Signaling Technology (CST), Danvers, MA, USA) containing a cOmplete Mini protease inhibitor cocktail and PhosSTOP phosphatase inhibitor cocktail (Sigma-Aldrich, St. Louis, MO, USA). The homogenates were then centrifuged at 10,000× *g* for 10 min at 4 °C, and the supernatants were collected. After determination of the protein concentrations, the samples were diluted in 3 × Blue Loading Buffer (CST) and denatured at 95 °C for 5 min. Then, 10 μg protein was separated on 7.5%, 10% or 12% TGX gels (Bio-Rad, Hercules, CA, USA), and subsequently transferred to polyvinylidene difluoride membranes. After the membranes were blocked with 5% skim milk in Tris-buffered saline with 0.1% Tween 20 (TBST) for 1 h at room temperature, they were incubated overnight at 4 °C with the following primary antibodies: phospho-Akt (Ser473, cat#9275, CST), total Akt (cat#4691, CST), phospho-mTOR (Ser2448, cat#2971, CST), total mTOR (cat#2983, CST), phospho-p70S6K (Thr389, cat#9234, CST), total p70S6K (cat#2708, CST), phospho-rpS6 (Ser240/244, cat#2215, CST), total rpS6 (cat#2217, CST), phospho-4EBP1 (Thr37/46, cat#9459, CST), total 4EBP1 (cat#9644, CST), phospho-eEF2 (Thr56, cat#2331, CST), total-eEF2 (cat#2332, CST), ubiquitinated proteins (cat#3936, CST), MuRF-1 (ab172479, Abcam, Cambridge, UK) and Atrogin-1 (ab168372, Abcam). The membranes were then incubated for 1 h with the appropriate secondary antibodies at room temperature and visualized using chemiluminescence (Immobilon Forte Western HRP substrate, Millipore, CA, USA). The bands were detected with an LAS 4000 system (GE Healthcare, Little Chalfont, UK). Ponceau S staining was performed to verify equal loading between lanes and for normalization. The band intensities were quantified with ImageJ software (National Institutes of Health, Bethesda, MD, USA).

### 2.6. RNA Extraction and Real-Time qPCR

The total RNA was extracted from muscle samples with ISOGEN II (Nippon Gene, Toyama, Japan) according to the manufacturer’s instructions. The RNA concentrations were determined using a NanoDrop 2000 (Thermo Fisher Scientific, Waltham, MA, USA), and 1.0 μg of total RNA was reverse-transcribed into cDNA using ReverTra Ace (Toyobo Co., Ltd., Osaka, Japan). The gene expression levels of MurF-1 (Hs00822397_m1), Atrogin-1 (Hs01041408_m1), FoxO3a (Hs00818121_m1), IL-6 (Hs00174131_m1), IL-1beta (Hs01555410_m1) and GAPDH (Hs02758991_g1) were quantified by TaqMan Gene Expression Assays with a 7500 Fast Sequence Detection System (Applied Biosystems, Waltham, MA, USA). The evaluation of mRNA was performed as an exploratory analysis based upon the results of protein kinetics. For confirmation of the evidence, the samples were reblinded to those assessing the outcome by staff not involved in the trial before analysis.

### 2.7. Statistical Analysis

Previous studies examining the effect of LEAA intake on muscle protein synthesis included *n* = 8 for each group. Referring to these reports and considering the possibility of dropout, we recruited 12 participants per group [25,26].

For comparison of the items in subject characteristics, a *t*-test was used. For the determination of the effect of exercise and LEAA intake on each variable, time-dependent changes against baseline values within each treatment group were evaluated using Dunnett’s test. For the analysis of the effect of LEAA supplementation, the significance of the differences between two treatment groups at each time point was evaluated by ANCOVA adjusted for baseline values. All values are expressed as the mean ± SD. Statistical significance was indicated by *p* < 0.05.

## 3. Results

### 3.1. Blood Insulin, Lactate and Glucose Concentrations

The time-dependent changes in blood insulin, lactate and glucose concentrations are shown in Table 3. In the placebo group, the blood glucose concentration increased between Ex and 0 min post-exercise in the placebo group (*p* < 0.001, peaking at 98.5 ± 6.7 mg/dL), and that in the LEAA group increased at Ex (*p* < 0.01, peaking at 98.6 ± 5.5 mg/dL). However, no significant difference was observed between the groups at each time point (Figure 2A). The blood insulin concentration in the placebo group increased between Ex and 30 min post-exercise (*p* < 0.05, peaking at 9.7 ± 5.6 μIU/mL). The blood insulin concentration in the LEAA group increased between Ex and 45 min post-exercise (*p* < 0.05, peaking at 15.2 ± 8.4 μIU/mL) and was also significantly higher than that of the placebo group at 0-90 min post-exercise (*p* < 0.05, Figure 2B). The blood lactate concentrations increased between Ex and approximately 45 min post-exercise (*p* < 0.01, peaking at 70.5 ± 17.6 mg/dL) in the placebo group, and that in the LEAA group increased between Ex and approximately 30 min post-exercise (*p* < 0.01, 72.0 ± 33.9 mg/dL). However, no significant difference was observed between the groups at each time point (Figure 2C).

### 3.2. Plasma BCAAs and Total EAA Concentrations

Time-dependent changes in the plasma BCAAs and total EAA concentrations are shown in Table 3. In the placebo group, the plasma leucine concentration decreased slightly between Ex and approximately 90 min post-exercise (*p* < 0.01, lowest value at 106.6 ± 11.7 μM). The plasma leucine concentration in the LEAA group increased between 0 and approximately 60 min post-exercise (*p* < 0.01, peaking at 339.5 ± 116.9 μM) and was also significantly higher than that in the placebo group at Ex-90 min post-exercise (*p* < 0.01, Figure 3A). The plasma isoleucine concentration in the placebo group decreased slightly between 0 and approximately 90 min post-exercise (*p* < 0.001, bottoming at 53.6 ± 4.9 μM). The plasma isoleucine concentration in the LEAA group increased between 0 and approximately 45 min post-exercise (*p* < 0.01, peaking at 115.0 ± 33.7 μM) and was significantly higher than that in the placebo group at Ex-90 min post-exercise (*p* < 0.05, Figure 3B). The plasma valine concentration in the placebo group decreased slightly between Ex and approximately 90 min post-exercise (*p* < 0.05, bottoming at 198.7 ± 22.4 μM). The plasma valine concentration in the LEAA group increased between 0 and approximately 45 min post-exercise (*p* < 0.05, peaking at 298.7 ± 50.6 μM) and was significantly higher than that in the placebo group at Ex-90 min post-exercise (*p* < 0.05, Figure 3C). The plasma total EAA concentration in the placebo group decreased slightly between 0 and approximately 90 min post-exercise (*p* < 0.001, lowest level at 892.5 ± 57.5 μM). The plasma EAA concentration in the LEAA group increased between 0 and approximately 60 min post-exercise (*p* < 0.01, peaking at 1404.4 ± 273.9 μM) and was significantly higher than that in the placebo group at Ex-90 min post-exercise (*p* < 0.05, Figure 3D).

### 3.3. mTORC1 Signaling

The time-dependent changes in mTORC1 signaling factors are shown in Table 4. In the placebo group, the phosphorylation of Akt^Ser473^ increased at 10 min post-exercise (*p* < 0.001), and no significant difference was observed between the groups at each time point (Figure 4A). The phosphorylation level of mTOR^Ser2448^ did not change from the pre-exercise level in the placebo group, however, in the LEAA group, this level was significantly higher than that in the placebo group at 10 and 90 min post-exercise (*p* = 0.012 and *p* = 0.035, respectively, Figure 4B). The placebo group showed a tendency toward increased phosphorylation of p70S6K^Thr389^ at 90 min post-exercise (*p* = 0.051), and the phosphorylation of p70S6K^Thr389^ in the LEAA group was significantly higher than that in the placebo group at 10 min post-exercise (*p* = 0.020, Figure 4C). Similarly, the phosphorylation of rpS6^Ser240/244^ showed a significant increase at 90 min post-exercise in the placebo group (*p* = 0.045), and, in the LEAA group, this phosphorylation showed an increased trend at 10 min and was significantly higher at 90 min post-exercise than in the placebo group (*p* = 0.061 and *p* = 0.045, respectively, Figure 4D). The placebo group showed a tendency toward decreased phosphorylation of 4EBP1^Thr37/46^ at 10 min post-exercise (*p* = 0.096), while in the LEAA group at 90 min post-exercise, this phosphorylation was significantly higher than that in the placebo group (*p* = 0.015, Figure 4E). The placebo group showed no significant time-dependent change in the phosphorylation of eEF2^Thr56^. Additionally, no significant difference was observed between the placebo and LEAA groups at 10 and 90 min post-exercise (Figure 4F).

### 3.4. Ubiquitin–Proteasome System-Related Factors

The time-dependent changes in the ubiquitin–proteasome system-related factors are shown in Table 4. The expression of ubiquitinated proteins and the MuRF-1 and Atrogin-1 proteins in the placebo group showed no significant change after resistance exercise. Additionally, no significant difference was observed between the placebo and LEAA groups at each time point (Figure 5A–C). However, the mRNA expression of MuRF-1 and Atrogin-1 increased in the placebo group at 90 min post-exercise (*p* < 0.001 and *p* = 0.024, respectively), but no significant difference between the groups was observed at each time point for either factor (Figure 5E,F). The FoxO3a mRNA expression was increased in the placebo group at 10 and 90 min post-exercise (*p* < 0.001), but no significant difference was observed at each time point between the groups (Figure 5G).

### 3.5. Inflammatory Cytokines

The time-dependent changes in the mRNA expression of IL-6 and IL-1β are shown in Table 4. The mRNA expression of both IL-6 and IL-1β increased at 90 min post-exercise in the placebo group (*p* < 0.001 and *p* < 0.01, respectively), but no significant difference was observed between the groups at each time point (Figure 6A,B).

## 4. Discussion

We investigated the effect of LEAA supplementation on the acute response to resistance exercise. As previously reported, a single bout of resistance exercise activated mTORC1 signaling, increased the expression of genes involved in the ubiquitin–proteasome system, and evoked inflammatory responses in the present study. Here, we further demonstrated that LEAA supplementation (i) augmented resistance exercise-induced activation of mTORC1 signaling, (ii) did not attenuate the increase in genes involved in the ubiquitin–proteasome system, and (iii) did not attenuate the increase in the mRNA expression of inflammatory cytokines induced by a single bout of resistance exercise. These observations suggest that LEAA supplementation augments the activation of the muscle anabolic response, but does not augment/attenuate the muscle catabolic response after resistance exercise.

The resistance exercise protocol composed of 10 reps in three sets at an intensity of 70% 1-RM is known to induce muscle hypertrophy when it is performed as training in men [41]. As expected, this exercise protocol also induced mTORC1 activation in the vastus lateralis muscle in the present study. In addition, the amount of LEAA taken in the present study was 5 g, the net content of leucine was 2 g, and the concentration of leucine in the plasma reached more than 300 μM. The threshold of the leucine concentration for muscle anabolism was approximately 250–300 μM, which stimulates muscle protein synthesis [42]. These facts indicated that the quantity of LEAAs used in the present study was sufficient to augment the effect of resistance exercise on the anabolic response.

Consistent with previous studies [17,31,43], a single bout of resistance exercise increased the MuRF-1 mRNA expression. Additionally, we observed the elevation of FoxO3a mRNA expression, which was not observed in a previous study that used a similar exercise protocol [31]. The explanation for this discrepancy is currently unclear; however, it may be due to differences in the contraction mode (concentric only vs. concentric and eccentric). This issue should be elucidated in a future study. We expected the inhibitory effect of LEAAs on the ubiquitin–proteasome system after resistance exercise. However, LEAA supplementation did not show any significant effect on MuRF-1, Atrogin-1, or FoxO3a in the present study. A previous study reported that an intake of 85 mg of BCAAs/kg body weight (45% leucine, 30% valine, and 25% isoleucine) prevented the increase in the MuRF-1 protein and Atrogin-1 mRNA levels 3 h after a single bout of resistance exercise composed of a single leg press [44]. In the same study, the participants ingested BCAAs five separate times, from the resting period to 45 min post-exercise, and the expressions of MuRF-1 and Atrogin-1 were observed at 3 h post-exercise. Therefore, a longer observation period might be needed to observe changes in the expression of MuRF-1 and Atrogin-1. Our results suggest that the intake of 5 g of LEAAs does not suppress the ubiquitin–proteasome system after a single bout of resistance exercise in young men within 90 min post-exercise.

Consistent with previous studies [29,30,31], a single bout of resistance exercise evoked acute inflammatory responses, as indicated by the increase in the mRNA expression of IL-1β and IL-6 in the present study. EAAs were reported to attenuate inflammatory responses after high-intensity exercise or training. Matsumoto et al. reported that BCAA supplementation (5 g valine, 10 g leucine and 5 g isoleucine, for each training day) attenuated the increase in plasma creatine kinase concentration after 3 days of running 12–40 km in humans [45]. Kato et al. reported that LEAA intake (1 g/kg, same composition as the present study, leucine intake was 0.12–0.16 g/day) attenuated muscle injury in rat skeletal muscle after 50 eccentric contractions [46]. As inflammation is one of the secondary phenomena of muscle injury, we expected an anti-inflammatory effect of LEAA supplementation. However, an anti-inflammatory effect of LEAAs was not observed in the present study. Recently, Rowlands et al. investigated the effect of protein–leucine–carbohydrate–fat ingestion (70/15/180/30 g, 23/5/180/30 g, or 0/0/274/30 g) on the regenerative inflammo-myogenic transcriptome in skeletal muscle following a 100 min cycle exercise [47]. In the previous study, transcriptome and bioinformatic analyses indicated the attenuation of IL-6 gene expression and predicted attenuation of IL-6 and IL-1β activation at 240 min after exercise, but not by 30 min after exercise [47]. These findings indicated that LEAAs have difficulty exerting anti-inflammatory effects in the early phase after a single bout of resistance exercise. Analysis of the mRNA expression of inflammatory cytokines at later phases after a single bout of resistance exercise would be useful to determine the presence/absence of an anti-inflammatory effect of LEAA supplementation.

Growing evidence demonstrated the effect of leucine and EAAs on the muscle anabolic response in vivo and in vitro [6,8,23,26,48]. In the present study, LEAA supplementation augmented the phosphorylation of mTOR^Ser2448^, p70S6K^Thr389^, rpS6^Ser240/244^, and 4EBP1^Thr37/46^ at 10 and/or 90 min post-exercise. Given the required time for the absorption and circulation of LEAAs, the enhancement of the phosphorylation of mTOR, p70S6K and rpS6 (trend) at 10 min post-exercise might be provided by the pre-exercise intake of LEAAs. The contribution of the early activation of mTORC1 to muscle anabolism in the later post-exercise period is currently unclear, and should be elucidated in future studies. To our knowledge, this is the first study to show the effect of low-dose LEAA intake on mTORC1 in young men. In previous human studies, the activation of muscle protein synthesis induced by resistance exercise or EAA ingestion was reported to be suppressed by rapamycin administration [5,49]. Additionally, some previous studies reported that LEAA supplementation increased muscle protein synthesis at the basal level, and after resistance exercise in older women [25,26]. These findings indicate that LEAA supplementation augments the muscle anabolic effect induced by resistance exercise in young men. Further studies demonstrating the effectiveness of LEAA supplementation on muscle hypertrophy induced by resistance exercise training would strengthen the evidence supporting the application of LEAA in various fields.

The present study has some potential limitations. First, we did not evaluate time-dependent changes in each factor after 90 min post-exercise. In particular, the increase in the mRNA expression of inflammatory cytokines was reported to continue until 24 h post-resistance exercise. Hence, LEAA supplementation might affect the increase in the mRNA expression of inflammatory cytokines beyond 90 min post-exercise in the present study. Second, we did not have a placebo group with a matched amount of nitrogen intake. Although the effectiveness of LEAAs on the mTORC1 signal response after resistance exercise was clarified by comparing the intake of LEAAs and maltitol in the present study, the LEAA-specific effect, compared to that of other amino acids, was not elucidated.

In conclusion, a single bout of resistance exercise evoked the activation of mTORC1 signaling, and increased the mRNA expression of ubiquitin–proteasome system-related factors and inflammatory cytokines in young men. LEAA supplementation did not reduce the increase in the mRNA expression of ubiquitin–proteasome system-related factors and inflammatory cytokines, but effectively augmented the activation of mTORC1 signaling induced by resistance exercise. These findings suggest that LEAA supplementation is useful for augmenting the effect of resistance exercise in young men. Further studies are required to assess the applicability of LEAAs in athletes, as well as patients in clinical settings.

## Figures and Tables

**Figure 1 nutrients-12-02421-f001:**
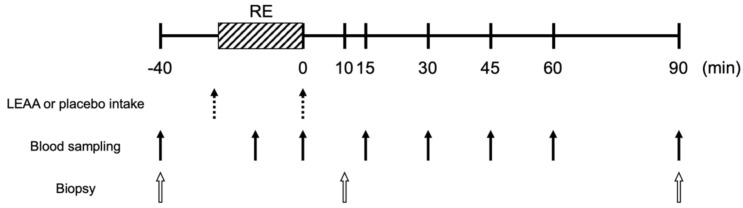
Schematic diagram of the experiment. Resistance exercise (RE) consists of 10 reps × three sets of leg extension and leg flexion at 70% of the one-repetition maximum. Subjects performed three sets of leg extension and then performed three sets of leg flexion. Blood sampling for the resistance exercise was taken during the switching of exercise events. Biopsies at rest and after exercise were taken from different legs.

**Figure 2 nutrients-12-02421-f002:**
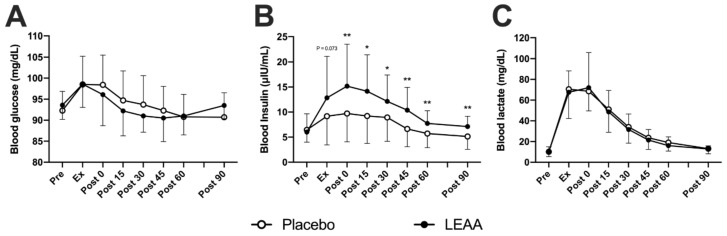
The effect of LEAA supplementation on blood glucose, insulin, and lactate concentrations after a single bout of resistance exercise. (**A**) Blood glucose, (**B**) blood insulin and (**C**) blood lactate. Data are expressed as the mean ± SD. The data were analyzed using ANCOVA adjusted for baseline values. * *p* < 0.05, ** *p* < 0.01 vs. the placebo group at the same time point.

**Figure 3 nutrients-12-02421-f003:**
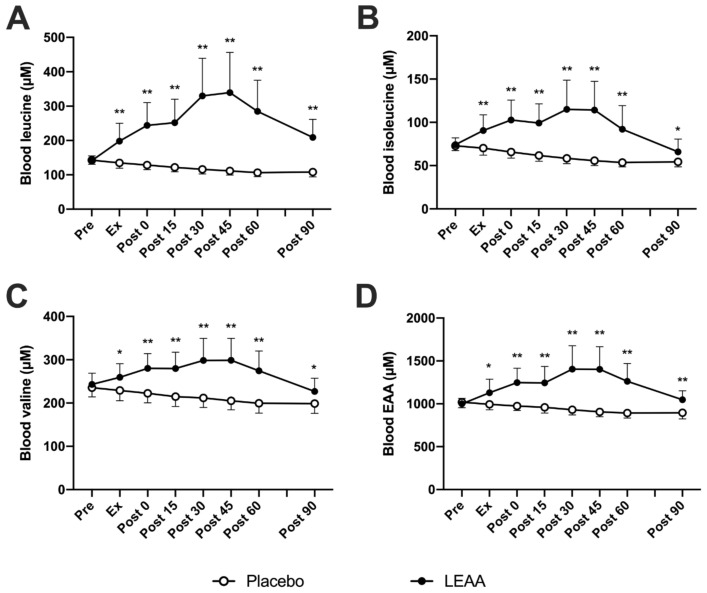
The effect of LEAA supplementation on amino acid concentrations after a single bout of resistance exercise. (**A**) Leucine, (**B**) isoleucine, (**C**) valine and (**D**) the total EAAs. The data were analyzed using ANCOVA adjusted for baseline values. The data are expressed as the mean ± SD. * *p* < 0.05, ** *p* < 0.01 vs. placebo group at the same time point.

**Figure 4 nutrients-12-02421-f004:**
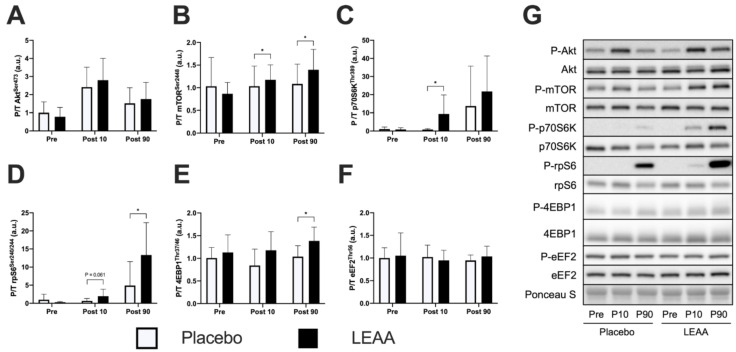
The phosphorylation of proteins in mTORC1 signaling. (**A**) Akt, (**B**) mTOR, (**C**) p70S6K, (**D**) rpS6, (**E**) 4EBP1, (**F**) eEF2 and (**G**) the representative bands. The data were analyzed using ANCOVA adjusted for baseline values. The data are expressed relative to the pretreatment condition in the placebo group as the mean ± SD. * *p* < 0.05 vs. the placebo group at the same time point.

**Figure 5 nutrients-12-02421-f005:**
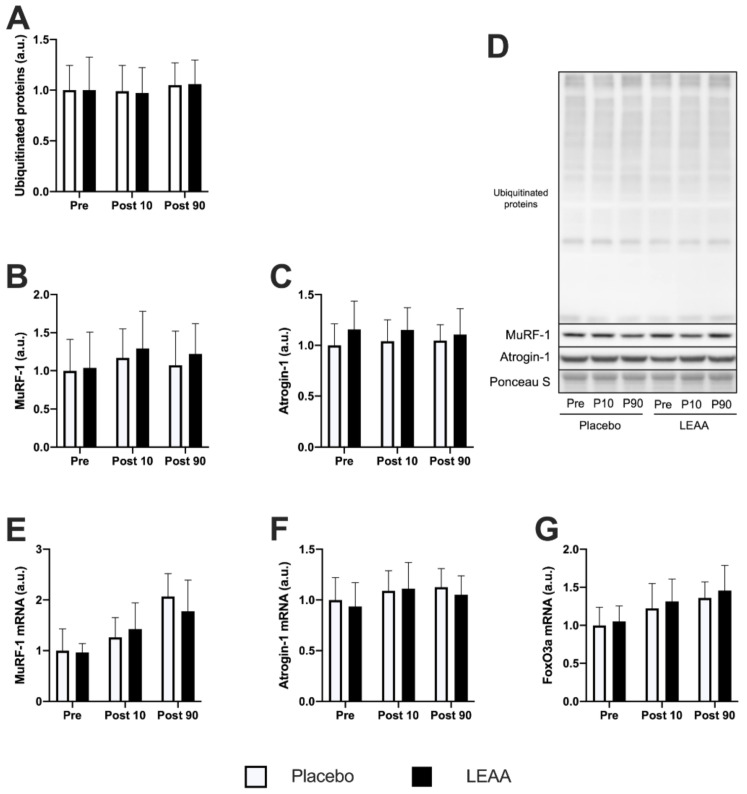
The protein and mRNA expression of factors in the ubiquitin–proteasome system. The expression of (**A**) ubiquitinated proteins, (**B**) MuRF-1, (**C**) Atrogin-1, and (**D**) the representative bands. The mRNA expression of (**E**) MuRF-1, (**F**) Atrogin-1, and (**G**) FoxO3a. The data were analyzed using ANCOVA adjusted for baseline values. The data are expressed relative to the pretreatment conditions in the placebo group as the mean ± SD. No significant difference was observed between the groups at each time point.

**Figure 6 nutrients-12-02421-f006:**
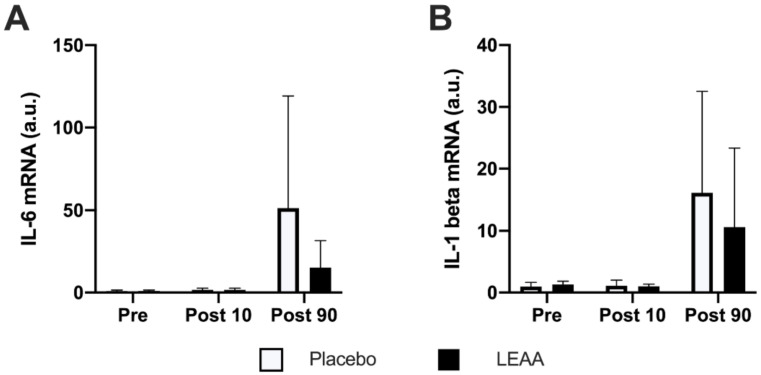
The mRNA expression of inflammatory cytokines. The mRNA expression of (**A**) IL-6 and (**B**) IL-1β. The data are expressed relative to the pretreatment condition in the placebo group as the mean ± SD. The data were analyzed using ANCOVA adjusted for baseline values. No significant difference was observed between the groups at each time point.

**Table 1 nutrients-12-02421-t001:** Subject characteristics.

	Placebo	Leucine-Enriched Essential Amino Acid (LEAA)	*p*-Value (*t*-Test)
Age (years)	21.4 ± 1.3	21.8 ± 1.5	0.535
Height (cm)	172.5 ± 6.5	171.8 ± 5.8	0.782
Weight (kg)	64.4 ± 9.3	61.4 ± 5.0	0.377
BMI (kg/m^2^)	21.5 ± 1.9	20.9 ± 1.9	0.443
**One-repetition maximum (1-RM)**			
Leg Extension (kg)	118.8 ± 27.5	126.1 ± 15.7	0.475
Leg Curl (kg)	84.7 ± 21.2	85.4 ± 10.1	0.926

Values are expressed as the mean ± SD. No significant difference was observed.

**Table 2 nutrients-12-02421-t002:** Composition of LEAAs and placebo taken per time.

	LEAA (g)	Placebo (g)
l-Leucine	1.00	
l-Isoleucine	0.27	
l-Valine	0.28	
l-Threonine	0.23	
l-Methionine	0.08	
l-Histidine hydrochloride	0.04	
l-Lysine hydrochloride	0.42	
l-Tryptophan	0.02	
l-Phenylalanine	0.17	
Maltitol	0.08	2.69
Excipient and Perfume	0.35	0.25
Total	2.94	2.94

Amino acids were replaced with maltitol in the placebo.

**Table 3 nutrients-12-02421-t003:** Time-dependent changes in the blood parameters.

Item	Placebo							LEAA						
	Ex	P 0	P 15	P 30	P 45	P 60	P 90	Ex	P 0	P 15	P 30	P 45	P 60	P 90
Glucose	↑ ***	↑ ***	→	→	→	→	→	↑ **	→	→	→	→	→	→
Insulin	↑ *	↑ **	↑ *	↑ *	→	→	→	↑ ***	↑ ***	↑ ***	↑ **	↑*	→	→
Lactate	↑ ***	↑ ***	↑ ***	↑ ***	↑ **	→	→	↑ ***	↑ ***	↑ ***	↑ **	→	→	→
Leucine	↓ **	↓ ***	↓ ***	↓ ***	↓ ***	↓ ***	↓ ***	→	↑ **	↑ ***	↑ ***	↑ ***	↑ ***	→
Isoleucine	→	↓ ***	↓ ***	↓ ***	↓ ***	↓ ***	↓ ***	→	↑ **	↑ *	↑ ***	↑ ***	→	→
Valine	↓ *	↓ ***	↓ ***	↓ ***	↓ ***	↓ ***	↓ ***	→	↑ *	↑ *	↑ ***	↑ ***	→	→
Total EAA	→	↓ ***	↓ ***	↓ ***	↓ ***	↓ ***	↓ ***	→	↑ ***	↑ **	↑ ***	↑ ***	↑ ***	→

Arrows pointing up and down indicate increases and decreases in the pretreatment values in each group. Horizontal arrows indicate unchanged values from the pretreatment values in each group. Ex and P 0-P 90 indicate sampling timepoints taken after the completion of leg extension and 0-90 min post-exercise, respectively. The data were analyzed using Dunnett’s test. * *p* < 0.05, ** *p* < 0.01, *** *p* < 0.001 vs. pretreatment for each group.

**Table 4 nutrients-12-02421-t004:** Time-dependent changes in muscle parameters.

	Placebo		LEAA	
	Post 10	Post 90	Post 10	Post 90
**Protein Content/Phosphorylation**				
P/T Akt	↑ ***	→	↑ ***	↑ **
P/T mTOR	→	→	↑ *	↑ ***
P/T p70S6K	→	↑ ^†^	→	↑ ***
P/T rpS6	→	↑ *	→	↑ ***
P/T 4EBP1	↓ ^†^	→	→	→
P/T eEF2	→	→	→	→
Ubiquitinated Proteins	→	→	→	→
MuRF-1	→	→	→	→
Atrogin-1	→	→	→	→
**mRNA Expression**				
*murf-1*	→	↑ ***	↑ **	↑ ***
*atrogin-1*	→	↑ *	↑ *	↑ ^†^
*foxo3a*	↑ **	↑ ***	↑ **	↑ ***
*il-1beta*	→	↑ ***	→	↑ **
*il-6*	→	↑ **	→	↑ **

Arrows pointing up and down indicate increases and decreases in the pretreatment values in each group. Horizontal arrows indicate no change from the pretreatment values in each group. The items written in italic indicate changes in the mRNA. The data were analyzed using Dunnett’s test. ^†^
*p* < 0.1, * *p* < 0.05, ** *p* < 0.01, *** *p* < 0.001 vs. the pretreatment values for each group.

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
