# Peer review of "The Effect of Leucine-Enriched Essential Amino Acid Supplementation on Anabolic and Catabolic Signaling in Human Skeletal Muscle after Acute Resistance Exercise: A Randomized, Double-Blind, Placebo-Controlled, Parallel-Group Comparison Trial"

_nutrients, 2020, doi:10.3390/nu12082421_

Round 1

Reviewer 1 Report

General comments:

The authors report on a small study investigating the effects of leucine enriched amino acid supplement on anabolic and catabolic signalling. The study may be of interest to readers.

Some general comments:

Introduction:

Line 58-59: However, BCAAs were reported to effectively stimulate mTORC1 signalling by promoting intake together with other EAAs [24]. - It is not clear what you mean here. This suggests intake of BCCA causes people to consume further AAs? This is unclear, perhaps you could re-phrase this sentence?

Table 1: 'Leg Carl' presumably this is a typo. Mass should be used in preference to weight. Units for BMI should be kg/m2 not kg/cm2

Method:

Line 101: What was the standard dinner? Was it fed in the lab or given to participants to consume at home? 

Were participants asked to refrain from intense physical activity in the days before the tests? 

Line: 104: Which leg were the biopsies taken from?

Line 107: what was the purpose of measuring RPE? 

Line 110 'is' should read 'was'

Line 111: What equipment was the exercise protocol performed on?

Line 112: How was the 1 RM determined?

Line 115: I am not sure what you mean by 'during the switch of exercise events' 

Fig 1 seems to indicate that a blood sample may have been collected between exercise sets. Was this the case? If so, it is not mentioned in the text of your methods section.

Line 119: What g were the blood samples centrifuged at?

How were your blood samples stored before analysis?

For all your analyses how many replicate samples were analysed? 

How did you determine your sample size?

You have performed Dunnett's tests to look at within group changes from baseline and seemingly multiple ANCOVA for each outcome variable at each time point?  Did you consider using a mixed model ANOVA with time points as your within group factor and treatment as a between group factor?

Results:

The tables that display whether outcomes have changed  within group from baseline using arrows are not very informative. The actual data is displayed more clearly in your figures which also present the between group comparisons (arguably the most important ones in a RCT). Perhaps the figures are adequate and the results section would be clearer without your tables containing the arrows?

Discussion: 

Line 317: creatine kinase 'activity' do you mean concentration?

Line 332:

"However, in contrast to these previous studies, an anti-inflammatory effect of LEAAs was not observed in the present study" - In the preceding sentences the studies you seem to be referring to have reported effects on muscle damage not inflammation.

In a previous study, attenuation of IL-1β and IL-6 expression was predicted by 240 min after exercise but not by 30 min after exercise. - This sentence is unclear. I assume you are referring to reference 46, but this is not clear from your wording. Also did they 'predict' attenuation of the cytokines or find an attenuation? 

Line 336: Perhaps you could justify your choice of two post exercise time points for the muscle biopsies?

It is not clear what you mean in the final sentence. 

Reviewer 2 Report

I would like to commend the authors for their engaging and interesting paper exploring leucine-enriched EAA supplementation following resistance exercise and I appreciate the opportunity to review the manuscript. Please see below for specific comments and suggestions on the text:

Abstract

Line 23-24: The use of ‘P < 0.05’ for each signalling target of interest is not very informative. I suggest that this be changed to the specific amino acid residue of each phosphorylation target. You may also include an approximation of the extent of phosphorylation increase (two-fold, etc).

Introduction

Line 40-41: please clarify this sentence “…..repeated exercise during training and accumulation of protein induce skeletal muscle hypertrophy”.   

Line 45: Space after ‘The…’ needs to be corrected.

Line 57: I suggest changing ‘more highly’ to ‘more robustly’ or ‘to a greater extent’.

Materials and Methods

Why was a parallel-group design implemented as opposed to a crossover design? 

Table 1: Exact P values to two decimal places (i.e., those greater than 0.01) and three decimal places (<0.01) would be more informative here to support between-group comparisons. ‘1RM’ would be preferable in bold font. Leg carl should be corrected to leg curl.

Line 101: can you specify the content of the standardised meal? Was daily protein intake controlled for?

Line 118: I suggest changing this to ‘Blood Insulin, Glucose, Lactate and Amino Acid Concentrations’

Table 2: The table footnote should be integrated into the caption ideally.

Line 168-173: Were statistical power calculations performed? If so please state these and if not provide some justification for the sample size chosen.

Results

Table 3 should not have RPE data. Please clarify abbreviations in table footnotes if possible.

The titles in bold above each figure should be removed and/or integrated within the Y-axis labels. This will improve the figure presentation in my view.

As mentioned above, the use of P<0.05 provides very limited information. Please provide exact P values where reasonable.

Line 220: should be ‘mTORC1 Signalling’

Line 222: remove ‘an effector of mTORC1’. Include this is discussion as necessary. Repeat for other instances in this paragraph.

Table 4: The caption should be above the table. I also suggest a heading in bold ‘Protein Content/Phosphorylation’ and another for ‘mRNA Expression’.

Figure 5: why are data expressed as relative to the pretreatment condition in the placebo group here when previous figures compare to pretreatment values in the respective conditions? My thoughts are that these factors should be compared to their corresponding baseline values within each group, and this would be consistent with the rest of the paper.

Discussion

Line 314: Does the increase in mRNA expression of IL-1b and IL-6 necessarily indicate acute inflammation? Levels of mRNA expression do not always align with protein abundance (https://genomebiology.biomedcentral.com/articles/10.1186/gb-2003-4-9-117). The absence of inflammatory cytokine protein data in the manuscript suggests that a more cautious stance should be taken here.

The authors acknowledge the lack of a nitrogen-balanced control/placebo and this is an important observation.

General Comments

The manuscript is concise and informative overall, with a large amount of data presented. However, I suggest a thorough edit as there are a number of spelling and grammatical errors. 
